# Antibody Response and Adverse Events of AZD1222 COVID-19 Vaccination in Patients Undergoing Dialysis: A Prospective Cohort Study

**DOI:** 10.3390/vaccines10091460

**Published:** 2022-09-03

**Authors:** Hsi-Hao Wang, Jia-Ling Wu, Min-Yu Chang, Hsin-Mian Wu, Li-Chun Ho, Po-Jui Chi, Ching-Fang Wu, Wan-Chia Lee, Hung-Hsiang Liou, Shih-Yuan Hung, Yi-Che Lee

**Affiliations:** 1Division of Nephrology, Department of Internal Medicine, E-DA Hospital, Kaohsiung 82445, Taiwan; 2School of Medicine, College of Medicine, I-Shou University, Kaohsiung 84001, Taiwan; 3Department of Medical Quality, E-DA Hospital, Kaohsiung 82445, Taiwan; 4Department of Public Health, College of Medicine, National Cheng Kung University, Tainan 70101, Taiwan; 5Department of Laboratory Medicine, E-DA Hospital, Kaohsiung 82445, Taiwan; 6Division of Nephrology, Department of Internal Medicine, E-DA Cancer Hospital, Kaohsiung 82445, Taiwan; 7Department of Rehabilitation, Kaohsiung Municipal United Hospital, Kaohsiung 80457, Taiwan; 8Division of Nephrology, Department of Internal Medicine, Hsin-Jen Hospital, New Taipei City 23561, Taiwan; 9Division of Nephrology, Department of Internal Medicine, E-DA Dachang Hospital, Kaohsiung 82445, Taiwan

**Keywords:** COVID-19, vaccination, AZD1222, antibody response, adverse events, dialysis, hemodialysis, peritoneal dialysis, cohort study

## Abstract

This study observed the antibody response and adverse events of AZD1222 (Oxford/AstraZeneca) vaccination in dialysis patients. A prospective cohort study was conducted in E-Da Healthcare Group hospitals between 1 July and 30 November 2021. Patients receiving hemodialysis (HD, *n* = 204) or peritoneal dialysis (PD, *n* = 116) were enrolled alongside healthy subjects (control, *n* = 34). Anti-SARS-CoV-2 S1 RBD IgG antibodies were measured before the first vaccination (T0), four to six weeks afterwards (T1), one week before the second dose (T2), and four to six weeks afterwards (T3). Adverse events were recorded one week after each dose. The positive IgG rates in the HD (T1: 72%; T2: 62%) and PD (T1: 69%; T2: 70%) groups were lower than the control group (T1: 97%; T2: 91%), with lower median antibody titers. At T3, the positive antibody response rates (HD: 94%; PD: 93%; control: 100%) and titers were similar. Titers were higher after the second dose in all groups. Adverse events were more severe after the first dose and less common with HD than PD or controls. Dialysis patients exhibited lower antibody responses than controls after the first dose of the AZD1222 vaccine but achieved similar responses after consecutive vaccination. Age, health status, two vaccine doses, and alcohol consumption may influence antibody levels.

## 1. Introduction

Coronavirus disease 2019 (COVID-19) is a highly contagious infectious disease caused by severe acute respiratory syndrome coronavirus 2 (SARS-CoV-2) [1] that has caused millions of deaths. End-stage renal disease (ESRD) patients experience 20–30 fold higher mortality and 3–4 fold more hospitalizations with SARS-CoV-2 infection compared with the general population [2]. This results in 20–30% mortality and 50% hospitalization rates [3,4], with many comorbidities and complications [5,6].

The need for regular treatment increases the risk of SARS-CoV-2 infection in dialysis patients because self-isolation and social distancing are difficult [4,7]. Emergency authorization of COVID-19 vaccines has led to calls for dialysis patients to be prioritized for vaccination [7,8]. However, this high-risk population was excluded from most COVID-19 vaccine trials [9]. These patients are likely to have an impaired immune response to COVID-19 vaccination [2,6,10] because of poor renal function, immunosuppression, and unusual exposures [5,6]. Therefore, collecting real-world data ensures these vaccines are as effective and safe as they are for other populations [11]. Evidence-based guidance also supports patients who may lack confidence in vaccine efficacy and have concerns about safety [12].

Currently, there are many COVID-19 vaccines available worldwide, including those based on mRNA, for example, BNT162b2 (Pfizer-BioNTech, New York,  NY, USA) and mRNA-1273 (Moderna, Cambridge, MA, USA), and the adenovirus-based AZD1222 (Oxford/AstraZeneca, Cambridge, UK) vaccine. AZD1222 is less expensive with easier storage than mRNA vaccines, making it popular, particularly in situations where maintaining cold storage is difficult [13]. However, extremely rare venous thromboembolic events with adenovirus vaccines often mean mRNA vaccines are preferred [14], despite mRNA vaccines also having rare side effects, including myocarditis [15]. Several studies have evaluated vaccination of dialysis patients with mRNA vaccines [2,16,17,18,19,20,21,22], most involving patients receiving hemodialysis (HD). These suggest that the humoral/antibody response is lower in dialysis patients than in healthy individuals [2,16,17], and a single dose may not elicit a response [12,23,24,25]. In contrast, studies on AZD1222 vaccination in dialysis patients [26,27,28,29] are scarce, and most importantly, studies on peritoneal dialysis (PD) patients are especially rare.

In response to an outbreak of a large-scale community infection of COVID-19 in Taiwan in May 2021, the government prioritized vaccination for high-risk groups. Specifically, dialysis patients were fully administered with the AZD1222 vaccine schedule by mid-June 2021 (the first dose) and mid-September 2021 (the second dose). Taking advantage of this opportunity, this study observed the antibody response and adverse events of AZD1222 vaccination in dialysis patients compared with healthy subjects.

## 2. Materials and Methods

### 2.1. Study Subjects

In this prospective cohort study, we recruited 402 participants from one would-be academic medical center and two district hospitals within the E-Da Healthcare Group (EDHG) in Kaohsiung, Taiwan, between 1 July and 30 November 2021. The dialysis group included patients with ESRD receiving HD (*n* = 222) or PD (*n* = 131). The control group (*n* = 49) consisted of non-dialysis subjects with less comorbid conditions, including nurses, laboratory staff, and volunteers from EDHG (Figure 1).

The dialysis group inclusion criteria were patients (1) aged over 20 years, (2) receiving regular outpatient dialysis treatment, and (3) having expressed consent to receive two doses of the AZD1222 COVID-19 vaccine as scheduled. The control group inclusion criteria were individuals (1) aged over 20 years, (2) with no history of severe chronic kidney disease (CKD stage 4 or 5), and (3) who had expressed consent to receive two doses of the AZD1222 vaccine as scheduled. The exclusion criteria for the groups were the same, including (1) persons diagnosed as having had a COVID-19 infection before or during the study, (2) those with a life expectancy of fewer than three months or with terminal cancer, (3) long-term use of immunosuppressants of more than three months or those undergoing chemotherapy, and (4) any person who received a vaccine other than the AZD1222 vaccine.

All study participants provided informed consent. This study was approved by the Institutional Review Board of the E-DA Hospital, Kaohsiung, Taiwan (EMRP-110-069).

### 2.2. Antibody Response

The Abbott AdviseDx SARS-CoV-2 IgG II assay [30] (analytical measuring interval (AMI): 22.0–25,000.0 AU/mL; Cut-off level for positive response: ≥50 AU/mL) was used to measure the serum concentrations of immunoglobulin G (IgG) against the receptor binding domain (RBD) of the SARS-CoV-2 S1 spike protein. Blood tests were collected at: T0, before the first dose of vaccine, (because Kaohsiung City, Taiwan, had low rates of COVID-19 infection during the study, this was only performed on some subjects to establish background values); T1, four to six weeks after the first dose of vaccine, (efforts were made to try to coordinate with routine blood tests to reduce the negative effects of the extra blood draw); T2, one week before the second dose (to establish baseline concentration); and T3, four to six weeks after the second dose (to assess the antibody response after both injections of the vaccine were complete) (Figure 2).

### 2.3. Adverse Events

We investigated adverse events (AE) one week after each vaccine dose through personal interviews with a questionnaire released by the Taiwan Society of Nephrology. The local AE included redness, swelling, pain, tenderness, and painkiller use. The systemic AE included fever, headache, muscle/joint pain, abdominal pain, diarrhea, anorexia/vomiting, fatigue, and skin rash/urticaria.

### 2.4. Baseline Characteristics

Baseline characteristics, including age, sex, body mass index (BMI), smoking and alcohol consumption (yes or no in the past year), dialysis vintage, etiology of ESRD, and comorbidities (history of diabetes mellitus (DM), hypertension, cardiovascular disease, cerebrovascular accident, chronic obstructive pulmonary disease, hepatitis, malignancy, autoimmune disease, or human immunodeficiency virus infection) were collected through personal interviews and/or search of medical records.

### 2.5. Statistical Analyses

Values are expressed as mean ± standard deviation (SD), median interquartile range (IQR) for continuous variables, or number (percentage) for categorical variables. Those that followed normality assumptions were evaluated using *t*-test and analysis of variance (ANOVA), and those that violated normality assumptions were assessed using the Wilcoxon rank sum test. The Fisher exact test was used for adjustment when the number of subjects was less than five. Chi-square and Kruskal–Wallis tests analyzed the antibody titer values for the groups at the T0, T1, T2, and T3 timepoints. Titers were compared between T1 and T3 using the Wilcoxon rank sum test. AE were compared using the chi-square test. Timeframe and factors/AE associated with log IgG levels were estimated using a multivariate linear mixed model. Age, sex, dialysis modality, and comorbidities were adjusted in the multivariate analysis to avoid confounding effects. All statistical analyses were conducted with SAS version 9.4 (SAS Institute, Cary, NC, USA) or R version 4.1.0. Graphs were depicted using Excel (Microsoft, Microsoft Corporation, One Microsoft Way, Redmond, WA, USA). *p*-values < 0.05 were considered statistically significant.

## 3. Results

### 3.1. Baseline Demographics

After exclusions, 204 HD patients, 116 PD patients, and 34 normal controls were included (Figure 1). The HD group (64 ± 11 years) was older than the PD group (56 ± 11 years) and the control group (51 ± 16 years). There were more males in the HD (67%) and PD (58%) groups, with more females in the control group (56%). The HD group’s BMI (25.10 ± 4.04 kg/m^2^) was higher than that of the control group (24.09 ± 2.90 kg/m^2^) and PD group (23.85 ± 4.3 kg/m^2^). The ESRD etiology showed more patients with systemic disease in the HD group (42%), but more patients with parenchymal renal disease in the PD group (54%). The proportions of DM, hypertension, and cerebrovascular accident in both dialysis groups were higher than in the control group (Table 1).

### 3.2. Antibody Response of Dialysis Patients and Controls at Different Time Points

The positive SARS-CoV-2 RBD IgG antibody rates in the HD (T1: 72.06%; T2: 62.25%) and PD (T1: 68.97%; T2: 69.83%) groups were significantly lower than in the control group (T1: 97.06%; T2: 91.18%) at T1 and T2 (*p* < 0.05), but there was no significant difference among the three groups (HD: 93.63%; PD: 93.10%; and Control: 100%) at T3 (Table 2). The SARS-CoV-2 RBD IgG antibody titer values in the HD (T1: 111.40 (40.65, 342.30); T2: 76.45 (26.80, 195.50) AU/mL) and PD (T1: 146.05 (29.35, 335.50); T2: 114.75 (23.15, 239.20) AU/mL) groups were significantly lower than in the control group (T1: 392.15 (266.40, 968.90); T2: 216.15 (145.10, 510.90) AU/mL) at T1 and T2 (*p* < 0.05), but there was no significant difference among the three groups at T3. The antibody titer values declined from T1 to T2 in all three groups (Table 2) (Figure 3) but were all significantly higher at T3 than T1 (*p* < 0.05) (Figure 3). These results imply that the antibody-positive rates and titers of the HD and PD patients were similar to controls only after 4–6 weeks of the second dose of the AZD1222 vaccine (T3).

### 3.3. Adverse Events after AZD1222 Vaccination among Dialysis Patients and Controls

Both the local and systemic AE were more severe after the first dose of the AZD1222 vaccine when compared with the second dose. HD patients were less likely to experience local and systemic AE compared with PD patients and controls. Pain, tenderness, and a need for painkiller use were local AE with significant differences between groups, while fatigue, fever, headache, muscle/joint pain, abdominal pain, and anorexia/vomiting were systemic AE with significant differences (Figure 4).

### 3.4. Timeframe and Factors/AE Associated with Log SARS-CoV-2 RBD IgG Levels

A multivariate linear mixed model adjusted for age, sex, BMI, smoking, drinking, dialysis modality, and comorbidities (Table 3) indicated that being younger, relatively healthy (non-dialysis), with two doses of AZD1222 vaccination, and without alcohol consumption was positively associated with IgG levels. Compared with T0, the IgG antibody levels were 120.23(10^2.08^)-fold, 89.13(10^1.95^)-fold, and 676.08(10^2.83^)-fold higher at T1, T2, and T3, respectively (*p* < 0.001). In addition, the IgG antibody levels decreased with age (0.98(10^−0.01^), *p* = 0.003) (Table 3). Compared with the control group, the IgG antibody levels were 0.53(10^−0.27^)-fold and 0.46(10^−0.33^)-fold lower in the HD (*p* = 0.007) and PD (*p* = 0.001) groups, respectively (Table 3). For participants with alcohol consumption, their IgG antibody levels were 0.42(10^−0.38^)-fold lower than those without drinking habits (*p* = 0.009) (Table 3). All adverse events, age, sex, BMI, smoking, drinking, dialysis modality, and comorbidities were adjusted by another multivariate linear mixed model (Table 4). Tenderness and skin rash/urticaria were negatively associated with the log SARS-CoV-2 RBD IgG levels. The IgG antibody levels significantly decreased with the occurrence of tenderness (0.692(10^−0.16^), *p* = 0.04) or skin rash/urticaria (0.326(10^−0.49^), *p* = 0.03) AE (Table 4).

## 4. Discussion

While dialysis patients are at high risk of SARS-CoV-2 infection and death, they may exhibit poor immune responses to COVID-19 vaccination. The evidence regarding the efficacy and safety of the AZD1222 vaccine in this population is limited. The aim of this study was to investigate the antibody response and adverse events of AZD1222 vaccination in dialysis patients compared with healthy subjects. We found that dialysis patients exhibited lower antibody responses than controls after the first dose of the AZD1222 vaccine but achieved similar responses after the second dose. Adverse events were more severe after the first dose and less common in HD patients than PD or controls. Younger, relatively healthy participants, with two vaccine doses, and with no alcohol consumption had higher antibody levels, but those with tenderness and skin rash/urticaria had lower antibody levels.

Our results show that the immune response was lower after a single dose of the vaccine in both dialysis groups compared with the control group. However, after the second dose of AZD1222 vaccination, response rates, and antibody titers were similar to the controls. It suggested that AZD1222 vaccination is an effective approach with at least two doses of the AZD1222 vaccine in dialysis patients. A Korean study also indicated that HD patients showed lower antibody titers and neutralizing antibody activities compared with healthy subjects following the first dose of the AZD1222 vaccine [31]. Our study showed PD patients presented with a similar phenomenon.

In 308 chronic dialysis patients (both HD and PD) that received AZD1222 vaccination [28], there was a delayed immune response after the first dose, with a 38% positive antibody response at 2 weeks and 66% at 10 weeks. Four weeks after the second dose, the rates increased to 94% [28]. Our results were similar; 72% of the HD group and 69% of the PD group had positive antibodies four to six weeks after the first dose, which increased to 94% in the HD group and 93% in the PD group four to six weeks after the second dose. Importantly, the antibody titers at T3 were similar to the controls. This compares favorably with the studies on mRNA vaccines showing lower antibody titers after a second dose [16,17,18]. However, a comparison of AZD1222 vaccination alongside mRNA vaccination in dialysis patients suggests that vaccine type does not influence response rates [26,27,29]. Overall, both vaccine types demonstrate higher antibody responses in dialysis patients than that found after hepatitis B or influenza vaccination, suggesting either approach will offer a good degree of protection [32]. Though, there may be a threshold effect, providing some concern for patients with lower titers. The vaccine rollout has generally been successful in preventing serious COVID-19 and hospitalizations in at-risk populations [33,34]. Importantly, hospitalizations for COVID-19 decreased in dialysis patients in a French study [35].

The safety of AZD1222 COVID-19 vaccination in dialysis patients is supported by the lack of serious AE in this study. Those AE that did occur were more severe after the first dose of the vaccine and were less common in the HD group. This was similar in other studies. With BNT162b2 vaccination, healthy subjects had a higher incidence and greater severity of AE than the dialysis group [17]. With mRNA-1273 vaccines, dialysis patients experienced fewer systemic vaccination-related AE [18]. However, in contrast to our results, the recipients of BNT162b2 [36] and mRNA-1273 [37] vaccines reported more local and systemic reactions after the second dose.

Longer-term observation would assess the requirements for a third vaccine dose. A review of COVID-19 vaccines in CKD and kidney transplant patients suggested that the low-level antibody response means booster doses are likely required [38]. In general, there is concern over a waning response to vaccination over time [39]. The humoral response substantially decreases, especially among men, persons 65 years or older, and persons with immunosuppression six months after the second dose of the BNT162b2 vaccine [40]. In dialysis patients, high responders had increased side effects after a third dose of the mRNA vaccine, but low responders reached a neutralizing level of titers [41]. We intend to follow up the dialysis patients of this study after a third dose of the AZD1222 vaccine, once the Taiwan government releases the third dose of the vaccine.

Changing between adenovirus and mRNA vaccine types may increase response rates. This was effective in an immunosuppressed kidney transplant recipient, who had a low antibody titer after the first and second dose of the BNT162b2 vaccine but a positive response after a booster of the AZD1222 vaccine [42]. In healthy volunteers, heterologous AZD1222/mRNA-1273 vaccination provided higher immunogenicity than homologous AZD1222 vaccination [43]. This approach may be particularly prudent as SARS-CoV-2 variants of concern arise, as even a general population vaccinated with two doses of AZD1222 or BNT162b2 required a booster of BNT162b2 or mRNA-1273 to increase protection against symptomatic disease caused by the omicron variant [44]. We plan to evaluate the effectiveness of heterologous COVID-19 vaccination for dialysis patients in the future.

As the main feature of lower immune function in patients with ESRD receiving dialysis is associated with the reduction of dendritic cells and a Th1/Th2 T-cell ratio imbalance, we must not forget the cellular immune response. This dysfunction reduces the number of activatable T cells in the body, while accelerating T-cell aging, resulting in poor responses of dialysis patients to some vaccines [45]. However, the impact of this on COVID-19 vaccination needs to be studied in detail because the Th1 cellular interferon (IFN)-γ secreting immune responses may protect against severe cases and new SARS-CoV-2 variants [46]. In most people, functional non-neutralizing antibody responses and T-cell responses are largely preserved against SARS-CoV-2 variants after vaccination [47]. However, the T-cell response after COVID-19 vaccination is decreased in dialysis patients [18,22]. With mRNA-1273 vaccination, IFN-γ production was significantly lower in dialysis patients when compared with controls [18].

Age, hemoglobin levels, and DM were previously identified as factors associated with AZD1222 vaccine response in dialysis patients [28]. Age, immunosuppression, and previous infection were related to antibody response in studies that included AZD1222 vaccinated dialysis patients [27,29]. Similarly, older age was related to a lower response for dialysis patients who received mRNA vaccination [16,17,22] and is related to vaccine response in other populations [16,39]. Patients receiving dialysis had a lower immune response compared with controls of similar age [21]. We identified age, being relatively healthy, and alcohol consumption as factors related to antibody titers. There is some evidence that alcohol consumption impacts both T cells and B cells, increasing the risk of infection and lowering the response to vaccination against agents such as hepatitis B and Bacillus Calmette-Guérin [48,49]. There also seemed to be some relationship with those who experienced tenderness and skin rash/urticaria having lower antibody levels. The influence of these factors on vaccine response may need more detailed investigation.

This study has some strengths and limitations. To our knowledge, this is the first study to compare the adverse events of AZD1222 COVID-19 vaccination among HD, PD, and healthy subjects. This is also the first study to investigate the impact of lifestyles on AZD1222 vaccine effectiveness, including drinking alcohol and cigarette smoking. While most studies have focused on mRNA vaccines, our study focused on an adenovirus-based vaccine. This study fills the gap in current knowledge about the effectiveness and safety of COVID-19 vaccination in dialysis patients. Our findings may boost subsequent COVID-19 prevention and vaccination policies. One limitation is that we cannot ensure that all the antibodies detected in the study subjects were derived from the immune response produced by the vaccine (Anti-Spike IgG) rather than after SARS-CoV-2 infection. As we know, HD patients have a higher SARS-CoV-2 infection risk than PD patients due to older age, relevant comorbidity, higher contamination risk from routine hospital-based treatment, and lack of self-isolation. On the other hand, we chose healthcare workers as a control group due to limited vaccine resources for the general population at that time in Taiwan, and healthcare workers also have a higher exposure risk of SARS-CoV-2 infection. Funding and testing reagent difficulties meant we did not detect IgG antibodies against the SARS-CoV-2 nucleocapsid (N) protein. However, although there was an outbreak in Taipei (Northern Taiwan) during the study period, the overall risk in Kaohsiung (Southern Taiwan) was relatively low, and there were no confirmed COVID-19 cases in dialysis patients and our control subjects in the three EDHG hospitals. Age, sex, dialysis modality, and comorbidities were adjusted in the multivariate analysis to avoid confounding effects. A larger study might have allowed the groups to be matched in terms of age, sex, and BMI to avoid the risk of bias.

## 5. Conclusions

In summary, the results of this study suggest that patients undergoing dialysis who receive two doses of the AZD1222 vaccine produce an effective antibody response. However, the initial response after one dose is lower than in control subjects. Age, health status, two vaccine doses, alcohol consumption, and tenderness and skin rash/urticaria AE were factors that influenced antibody levels. AE were more severe after the first vaccine dose and were less common in HD patients than PD or controls.

## Figures and Tables

**Figure 1 vaccines-10-01460-f001:**
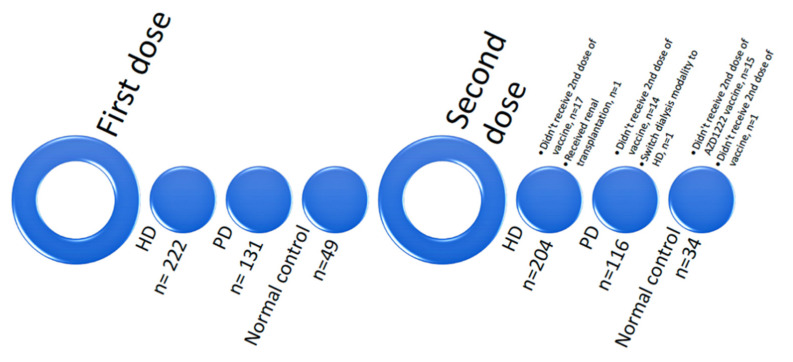
Study population.

**Figure 2 vaccines-10-01460-f002:**
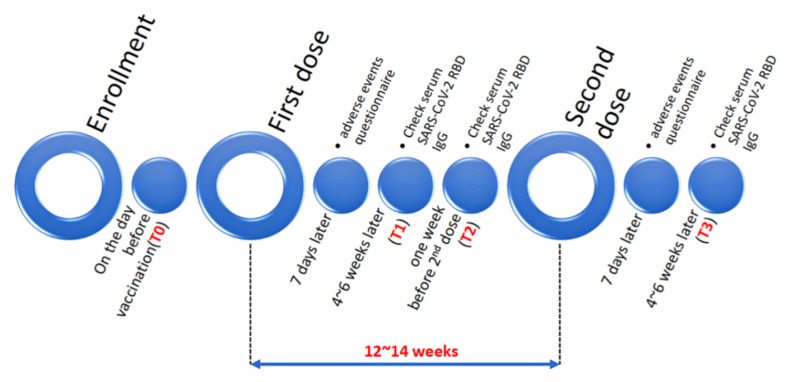
Study protocol.

**Figure 3 vaccines-10-01460-f003:**
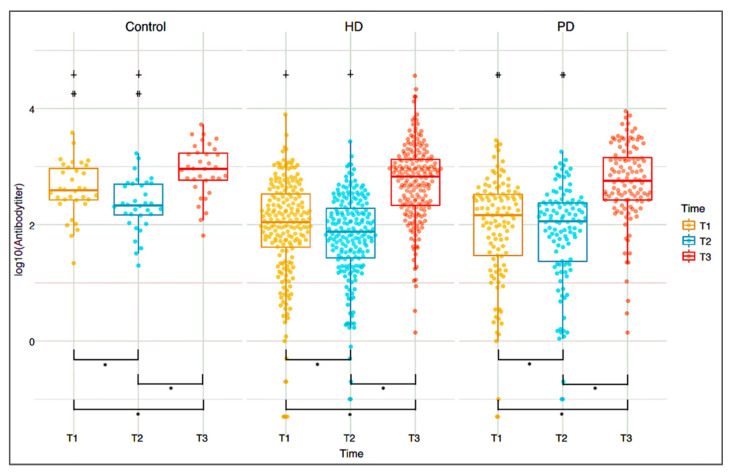
SARS-CoV-2 RBD IgG antibody titer values by group and time (T1 vs. T2 vs. T3). ^∔^ HD vs. Control (*p*-value < 0.05), ^⧺^ PD vs. Control (*p*-value < 0.05), * *p*-value < 0.05.

**Figure 4 vaccines-10-01460-f004:**
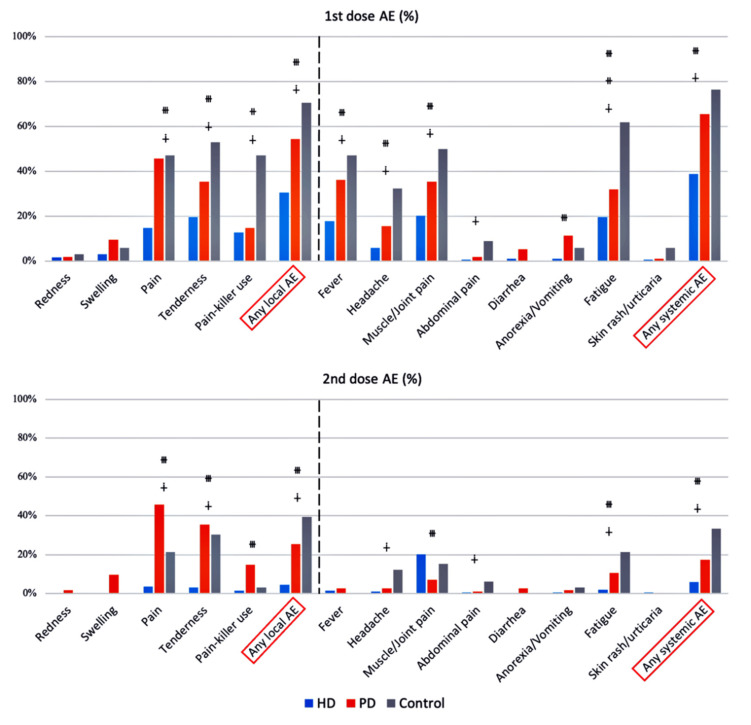
Local (on the left side of dotted line) and systemic (on the right side of dotted line) adverse events (AE) after the first and second vaccination with AZD1222. ^∔^ HD vs. Control, *p*-value < 0.05, ^⧺^ PD vs. Control, *p*-value < 0.05, ^⧻^ HD vs. PD, *p*-value < 0.05.

**Table 1 vaccines-10-01460-t001:** Baseline demographic data and characteristics of the subjects in the study.

Variables	HD	PD	Control	*p*-Value
	N = 204	N = 116	N = 34	
Age, mean (SD), year	64.21 ± 11.29 ^∔,^^⧻^	56.01 ± 11.12 ^⧻^	50.83 ± 15.96 ^∔^	<0.001 ***
Sex				0.022 *
Men	136 (67.00) ^∔^	67 (57.76)	15 (44.12) ^∔^	
Women	67 (33.00)	49 (42.24)	19 (55.88)	
BMI (kg/m^2^)	25.10 ± 4.04 ^⧻^	23.85 ± 4.3 ^⧻^	24.09 ± 2.90	0.023 *
Smoking	19 (9.31)	8 (6.90)	0 (0.00)	0.156 *^a^*
Drinking	8 (3.92)	2 (1.72)	3 (8.82)	0.127 *^a^*
Dialysis vintage, mean (SD), month	69.29 ± 64.33	58.99 ± 47.30	-	0.133
Etiology of ESRD				<0.001 ***
Parenchymal renal disease	88 (43.14)	63 (54.31)	-	
Systemic disease	86 (42.16)	44 (37.93)	-	
Obstructive uropathy and other disease of urinary system	4 (1.96)	1 (0.86)	-	
Polycystic kidney disease	8 (3.92)	3 (2.59)	-	
Others	18 (8.82)	5 (4.31)	-	
Comorbid conditions				
DM	108 (52.94) ^⧻^	38 (33.04) ^⧺,^^⧻^	3 (8.82) ^⧺^	<0.001 ***
HTN	164 (80.39)	91 (79.13) ^⧺^	8 (23.53) ^⧺^	<0.001 ***
CVD	46 (22.55)	24 (20.69)	2 (5.88)	0.082
CVA	17 (8.33) ^⧻^	1 (0.86) ^⧻^	0 (0.00)	0.005 **
COPD	7 (3.43)	0 (0.00)	0 (0.00)	0.075 *^a^*
Hepatitis	36 (17.65)	24 (20.69)	5 (14.71)	0.673
Malignancy	28 (13.73)	17 (14.66)	1 (2.94)	0.181
Autoimmune disease	2 (0.98)	3 (2.59)	1 (2.94)	0.336 *^a^*
HIV	4 (1.13)	0 (0.00)	0 (0.00)	0.407 *^a^*

Values are expressed as mean ± standard deviation for continuous variables or number (percentage) for the categorical variables. The *^a^* symbol indicates the Fisher exact test, which is used for adjustment. * *p*-value < 0.05, ** *p*-value < 0.01, *** *p*-value < 0.001. ^∔^ HD versus Control, *p*-value < 0.05, ^⧺^ PD versus Control, *p*-value <0.05, ^⧻^ HD versus PD, *p*-value < 0.05. Parenchymal renal disease: chronic glomerulonephritis, tubulointerstitial nephritis, chronic interstitial nephritis, IgA nephropathy, focal segmental glomerulonephritis, Chinese herb nephropathy, chronic pyelonephritis, acute kidney injury (no recovery). Systemic disease: nephrosclerosis, diabetes mellitus, malignant hypertension, gouty nephropathy, systemic lupus nephritis. Abbreviations: HD: hemodialysis; PD: peritoneal dialysis; SD: standard deviation; BMI: Body Mass Index; ESRD: end-stage renal disease; DM: diabetes mellitus; HTN: hypertension; CVD: cardiovascular disease; CVA: cerebrovascular accident; HIV: human immunodeficiency virus.

**Table 2 vaccines-10-01460-t002:** SARS-CoV-2 RBD IgG antibody titer values by group and time.

	T0	T1	T2	T3
HD (N = 204)	(*n* = 15)	(*n* = 204)	(*n* = 204)	(*n* = 204)
Days between two dates Median (IQR)		33 (33, 35)	75 (75, 77)	138 (138, 140)
Positive rate (%)	0	72.06 ^∔^	62.25 ^∔^	93.63
Median (IQR)	0.8 (0.00, 4.10)	111.40 (40.65, 342.30) ^∔^	76.45 (26.80, 195.50) ^∔^	677.0 (214.1, 1348.7)
PD (N = 116)	(*n* = 3)	(*n* = 116)	(*n* = 116)	(*n* = 116)
Days between two dates Median (IQR)		41 (39, 42)	74 (70, 76)	133 (132, 137)
Positive rate (%)	0	68.97 ^⧺^	69.83 ^⧺^	93.10
Median (IQR)	2.2 (0.00, 8.50)	146.05 (29.35, 335.50) ^⧺^	114.75 (23.15, 239.20) ^⧺^	573.55 (263.90, 1482.80)
Control (N = 34)	(*n* = 0)	(*n* = 34)	(*n* = 34)	(*n* = 34)
Days between two dates Median (IQR)		33 (29, 35)	69 (68, 69)	128 (108, 147)
Positive rate (%)		97.06 ^∔,^^⧺^	91.18 ^∔,^^⧺^	100.00
Median (IQR)		392.15 (266.40, 968.90) ^∔,^^⧺^	216.15 (145.10, 510.90) ^∔,^^⧺^	924.0 (580.6, 1741.5)

T0: Blood sampled on the day before the first dose of COVID-19 vaccination. T1: Blood sampled 4~6 weeks after the first dose of COVID-19 vaccination. T2: Blood sampled 1 week before the second dose of COVID-19 vaccination. T3: Blood sampled 4~6 weeks after the second dose of COVID-19 vaccination. ^∔^ HD vs. Control (*p*-value < 0.05), ^⧺^ PD vs. Control (*p*-value < 0.05). Abbreviations: SARS-CoV-2 RBD IgG: severe acute respiratory syndrome coronavirus 2 receptor binding domain immunoglobulin G; HD: hemodialysis; PD: peritoneal dialysis; IQR: interquartile range.

**Table 3 vaccines-10-01460-t003:** Timeframe and factors associated with log SARS-CoV-2 RBD IgG levels adjusted for age, sex, BMI, smoking, drinking, dialysis modality, and comorbidities.

Parameter	Estimate (Standard Error)	*p*-Value
Time (ref = T0)		
T1	2.08 (0.04)	<0.001 ***
T2	1.95 (0.04)	<0.001 ***
T3	2.83 (0.04)	<0.001 ***
Age, y	−0.01 (0.002)	0.003 **
Group (ref = Control)		
HD	−0.27 (0.10)	0.007 **
PD	−0.33 (0.10)	0.001 **
Drinking	−0.38 (0.14)	0.009 **

T0: Blood sampled on the day before the first dose of COVID-19 vaccination. T1: Blood sampled 4~6 weeks after the first dose of COVID-19 vaccination. T2: Blood sampled 1 week before the second dose of COVID-19 vaccination. T3: Blood sampled 4~6 weeks after the second dose of COVID-19 vaccination. ** *p*-value < 0.01, *** *p*-value < 0.001. Abbreviations: SARS-CoV-2 RBD IgG: severe acute respiratory syndrome coronavirus 2 receptor binding domain immunoglobulin G; HD: hemodialysis; PD: peritoneal dialysis.

**Table 4 vaccines-10-01460-t004:** Timeframe and adverse events associated with log SARS-CoV-2 RBD IgG levels, adjusted for age, sex, BMI, smoking, drinking, dialysis modality, and comorbidities.

Parameter	Estimate (Standard Error)	*p*-Value
Time (ref = T0)		
T1	2.08 (0.03)	<0.001 ***
T2	1.94 (0.04)	<0.001 ***
T3	2.83 (0.05)	<0.001 ***
Age, y	−0.01 (0.002)	0.001 **
Tenderness	−0.16 (0.08)	0.040 *
Skin rash/urticaria	−0.49 (0.34)	0.032 *

T0: Blood sampled on the day before the first dose of COVID-19 vaccination. T1: Blood sampled 4~6 weeks after the first dose of COVID-19 vaccination. T2: Blood sampled 1 week before the second dose of COVID-19 vaccination. T3: Blood sampled 4~6 weeks after the second dose of COVID-19 vaccination. * *p*-value < 0.05, ** *p*-value < 0.01, *** *p*-value < 0.001. Abbreviations: SARS-CoV-2 RBD IgG: severe acute respiratory syndrome coronavirus 2 receptor binding domain immunoglobulin G; HD: hemodialysis; PD: peritoneal dialysis.

## Data Availability

Not applicable.

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
