# Peer review of "Antibody Response and Adverse Events of AZD1222 COVID-19 Vaccination in Patients Undergoing Dialysis: A Prospective Cohort Study"

_vaccines, 2022, doi:10.3390/vaccines10091460_

Round 1

Reviewer 1 Report

The COVID-19 vaccination is a key element of the current worldwide anti-SARS-CoV-2 strategy. It is estimated that to acquire the herd immunity, we need to vaccinate approximately 70-80% of worldwide population. Currently, there are many different COVID-19 vaccines, which were approved by the FDA, EMA or other authorities, based on the data presented by manufacturers. However, the vaccines were tested in the clinical trials mostly on healthy volunteers, and do not include patients with specific comorbidities. Wang and colleagues in their manuscript analyzed the antibody response and adverse effect of AZD1222 (Astra Zeneca/Oxford University) vaccine among dialysis patients. The obtained results are of interests for scientists to understand the vaccine effect on the patient organism, as well as for healthcare authorities to plan the vaccination campaigns. The paper is relatively well-written, and the data are clear. But I have some comments (please see below), which in my opinion, may strengthen the manuscript.

MAJOR COMMENTS

1.     Please use the scientific name of the Astra Zeneca/Oxford University vaccine, i.e. AZD1222 or ChAdOx1. Please avoid the commercial name, i.e. Vaxzevria. Additionally, please be consistent with the nomenclature of the described vaccines.

2.     In my opinion, it would be beneficial to include the 95%CI in the Table 1.

3.     I would like to ask why authors decided to show on Figure 3 only data from T1 and T2. Why did authors not demonstrate the antibody titers on T3? I would strongly recommend to show the antibody titers in all tested time points.

4.     I am not sure why Figure 4 is not described in the tex. Please include in the Result section a description of analyzed adverse effects.

5.     Within Table 3 and 4 authors showed few factors associated with high antibody titers. How about the other factors, which were collected i.e. in the questionaries?

6.     Within the control group, authors decided to include the healthcare workers, which are very specific group due to the high SARS-CoV-2 infection risk. Please describe your choice in the Discussion.

7.     I would also recommend to add in the Discussion that the PD patients are in general in better condition and at lower SARS-CoV-2 infection risk in comparison to the HD.

MINOR COMMENTS

a.     Title – please use the scientific name of vaccine. In addition, please be consistent with the nomenclature, i.e. in the title authors wrote ‘side effects’ and later in the text ‘adverse effect’.

b.     Page 1 Lines 35-36. The sentences are not understood, definitely something is lacking.

c.     Page 1 Line 39 – please do not use ‘inoculation’ regarding the vaccination. You can inoculate your LB medium with bacteria, but not human with vaccine.

d.     Page 2 Line 85 – please explain all abbreviations, i.e. ESRD

e.     Page 2 Line 86 – what do authors understand as a ‘relatively healthy’. I would avoid such definitions. Please describe the control group according to the Table 1.

f.      Page 4 Line 124 – why do authors include ‘painkiller use’ as an adverse effect?

g.     Page 6 Figure 3 – please increase the font on the Figures, since in the current version it is really hard to read

h.     Page 7 Table 2 – authors decided to show the positive rate as %, but in the group column there is probably number of positive study participant and the positive rate is shown in the brackets. Please correct.

Author Response

Dear Editors,

My colleagues and I are very grateful for all the constructive comments from the two anonymous reviewers. Would you please kindly express our sincere appreciation to both of them? Without their efforts and careful advice, this manuscript would not have been improved so much. If the revised manuscript still needs to be improved, please kindly do not hesitate to let us know. Thank you very much for all your attention and efforts.

Sincerely yours,

Yi-Che Lee, (representing all co-authors)

Reviewer 1

MAJOR COMMENTS

1. Please use the scientific name of the Astra Zeneca/Oxford University vaccine, i.e. AZD1222 or ChAdOx1. Please avoid the commercial name, i.e. Vaxzevria. Additionally, please be consistent with the nomenclature of the described vaccines.

Thanks for your valuable comment. We have replaced Oxford/AstraZeneca (AZ) vaccine with AZD1222, Pfizer-BioNTech vaccine with BNT162b2, and Moderna vaccine with mRNA-1273 in the whole manuscript.

2. In my opinion, it would be beneficial to include the 95%CI in the Table 1.

Thanks for your comment, but our colleagues are not so sure which item or factor should include 95% CI in the table 1. Would you please let us know which data should include 95% CI in the table 1? My colleagues and I will be glad to show it.

3. I would like to ask why authors decided to show on Figure 3 only data from T1 and T2. Why did authors not demonstrate the antibody titers on T3? I would strongly recommend to show the antibody titers in all tested time points.

Thanks for your comment. In original version we showed T1 and “T3” antibody titers on Figure 3 to demonstrate the titers change after the first dose and the second dose of AZD1222 vaccination. In this revised version, we have shown T1, T2, and T3 antibody titers on figure 3 and it will be clearer for our readers to understand the serial titers change during the observation period.

4. I am not sure why Figure 4 is not described in the tex. Please include in the Result section a description of analyzed adverse effects.

Thanks for your kind reminder. The description of analyzed adverse effects was in Page 8 Lines 1-8.

5. Within Table 3 and 4 authors showed few factors associated with high antibody titers. How about the other factors, which were collected i.e. in the questionaries?

Thanks for your comment. In Table 3, we have adjusted for age, sex, BMI, smoking, drinking, dialysis modality, and comorbidities. In Table 4, we have adjusted all adverse events and same factors in Table 3. The factors we didn’t show on table 3 and Table 4 because they didn’t reach significant statistic results.

6. Within the control group, authors decided to include the healthcare workers, which are very specific group due to the high SARS-CoV-2 infection risk. Please describe your choice in the Discussion.

Thanks for your constructive comment. We have added on sentences as follows in our discussion:

Page 11 Lines 156-158~ Page 12 Line 1

On the other hand, we chose healthcare workers as control group due to limited vaccine resources for general population at that time in Taiwan and healthcare workers also have higher exposure risk of SARS-CoV-2 infection.

7. I would also recommend to add in the Discussion that the PD patients are in general in better condition and at lower SARS-CoV-2 infection risk in comparison to the HD.

Thanks for your valuable comment. We have added on sentences as follows in our discussion:

Page 11 Lines 153-156

As we know, HD patients have higher SARS-CoV-2 infection risk than PD patients due to older age, relevant comorbidity, higher contamination risk from routine hospital-based treatment and lack of self-isolation.

MINOR COMMENTS

a. Title – please use the scientific name of vaccine. In addition, please be consistent with the nomenclature, i.e. in the title authors wrote ‘side effects’ and later in the text ‘adverse effect’.

Thanks for your suggestion. We have revised the title and our manuscript as follows:

Antibody Response and Adverse Events of AZD1222 COVID-19 Vaccination in Patients Undergoing Dialysis: A Prospective Cohort Study

Page 2 Lines 86-88

Taking advantage of this opportunity, this study observed the antibody response and adverse events of AZD1222 vaccination in dialysis patients compared to healthy subjects.

Page 4 Line133

2.3. Adverse Events

b. Page 1 Lines 35-36. The sentences are not understood, definitely something is lacking.

Thanks for your kind reminder. We have revised the abstract as follows:

Page 1 Lines 35-36

At T3, the positive antibody response rates (HD: 94%; PD: 93%; control: 100%) and titers were similar.

c. Page 1 Line 39 – please do not use ‘inoculation’ regarding the vaccination. You can inoculate your LB medium with bacteria, but not human with vaccine.

Thanks for your kind reminder. We have revised the abstract as follows:

Page 1 Lines 37-39

Dialysis patients exhibited lower antibody responses than controls after the first dose of AZD1222 vaccine but achieved similar responses after consecutive vaccination.

d. Page 2 Line 85 – please explain all abbreviations, i.e. ESRD

Thanks for your kind reminder. We have checked and explained all abbreviations again. ESRD was explained in Page 2 Line 54.

e. Page 2 Line 86 – what do authors understand as a ‘relatively healthy’. I would avoid such definitions. Please describe the control group according to the Table 1.

Thanks for your suggestion. We have revised the manuscript as follows:

Page 2 Lines 94-96

The control group (n=49) consisted of non-dialysis subjects with less comorbid conditions, including nurses, laboratory staff, and volunteers from EDHG (Figure 1).

f. Page 4 Line 124 – why do authors include ‘painkiller use’ as an adverse effect?

Thanks for your comment. We used a questionnaire released by the Taiwan Society of Nephrology to record the adverse effects of AZD1222 vaccine on all dialysis patients in Taiwan. “Painkiller use” is one of the items classified as local adverse effect (redness, swelling, pain, tenderness, and painkiller use). Each item in this questionnaire was graded from 0 to 4 (i.e. from no discomfort to need emergent medical treatment). Painkiller use may downgrade the other local adverse events, so it needs to be noted. The grade of painkiller use was 0: no painkiller use, 1: prophylaxis use, 2: need oral painkiller use once per day, 3: need oral painkiller use more than once per day, 4: need intravenous painkiller use or emergent medical treatment. In this manuscript, we simplified the data and only extract “yes” or “no” results of each adverse effect.       

g. Page 6 Figure 3 – please increase the font on the Figures, since in the current version it is really hard to read

Thanks for your good suggestion. We have increased the font size and added on T2 data in Page 6 Figure 3.

h. Page 7 Table 2 – authors decided to show the positive rate as %, but in the group column there is probably number of positive study participant and the positive rate is shown in the brackets. Please correct.

Thanks for your kind reminder. We have removed the number and brackets of the positive rate in Page 7 Table 2.

You can also see the attachment.

Reviewer 2 Report

the manuscript describes the antibody response to the Oxford/AstraZeneca COVID-19 vaccine in dialysis patients. anti RBD IgG levels were measured at 3 time points after vaccination in HD PD and a contrrol group.

materials and methods:

could you please specify the vaccination schedule 

please specify the antibody cut-off level for positive response

results:

figure 4:

you may consider to change the bars filling in order to create better graphic separation between the groups.

you mention drinking habits - it is not clear how the population was classified according to this parameter.

table 3:

I belive you present a multivariate or adjusted analysis. if so could you mention the adjustment in the headline

table 4:

please specify if all AE are included or only systemic AE?

study design:

I wonder if you have results  on a more extended time of follow-up as antibody levels are expected to drop over a relatively short period

in case such information could be added to the manuscript it could generate more interest for the readers.

Author Response

Dear Editors,

My colleagues and I are very grateful for all the constructive comments from the two anonymous reviewers. Would you please kindly express our sincere appreciation to both of them? Without their efforts and careful advice, this manuscript would not have been improved so much. If the revised manuscript still needs to be improved, please kindly do not hesitate to let us know. Thank you very much for all your attention and efforts.

Sincerely yours,

Yi-Che Lee, (representing all co-authors)

Reviewer 2

materials and methods:

could you please specify the vaccination schedule 

Thanks for your comment. We have revised the introduction and Figure 2 to specify the vaccination schedule.

Page 2 Lines 84-86

Specifically, dialysis patients were fully administered with the AZD1222 vaccine schedule by mid-June 2021 (the first dose), and mid-September 2021 (the second dose).

Page 4 Figure 2

please specify the antibody cut-off level for positive response

Thanks for your comment. We have revised the manuscript as follows:

Page 3 Lines 119-122

The Abbott AdviseDx SARS-CoV-2 IgG II assay [31] (Analytical Measuring Interval (AMI): 22.0-25000.0 AU/mL; Cut-off level for positive response: ≥50 AU/mL) was used to measure the serum concentrations of immunoglobulin G (IgG) against the receptor binding domain (RBD) of the SARS-CoV-2 S1 spike protein.

results:

figure 4:

you may consider to change the bars filling in order to create better graphic separation between the groups.

Thanks for your comment. We have revised figure 4 and increased the font size as follows:

Page 8 Figure 4

you mention drinking habits - it is not clear how the population was classified according to this parameter.

Thanks for your comment. The alcohol consumption was collected by personal interview and the criteria was based on National Health Interview Survey in Taiwan. We asked our patients and control subjects if they had alcohol consumption in the past one year. So, we have revised our manuscript as follows:

Page 4 Lines 146-151

Baseline characteristics, including age, sex, body mass index (BMI), smoking and alcohol consumption (yes or no in the past one year), dialysis vintage, etiology of ESRD, and comorbidities (history of diabetes mellitus (DM), hypertension, cardiovascular disease, cerebrovascular accident, chronic obstructive pulmonary disease, hepatitis, malignancy, autoimmune disease, or human immunodeficiency virus infection) were collected through personal interviews and/or search of medical records.

table 3:

I belive you present a multivariate or adjusted analysis. if so could you mention the adjustment in the headline

Thanks for your valuable comment. We have added on the adjustment in the head line and revised the manuscript as follows:

Page 8 Lines 14-16

A multivariate linear mixed model adjusted for age, sex, BMI, smoking, drinking, dialysis modality, and comorbidities (Table 3) indicated that being younger, relatively healthy (non-dialysis), with 2 doses of AZD122 vaccination, and without alcohol consumption was positively associated with IgG levels.

Page 9 Table 3 headline

table 4:

please specify if all AE are included or only systemic AE?

Thanks for your comment. All AE are included in the model. We have revised Table 4 and our manuscript as follows:

Page 8 Lines 23-24

All adverse events, age, sex, BMI, smoking, drinking, dialysis modality, and comorbidities were adjusted by another multivariate linear mixed model (Table 4).

Page 9 Table 4 headline

study design:

I wonder if you have results on a more extended time of follow-up as antibody levels are expected to drop over a relatively short period in case such information could be added to the manuscript it could generate more interest for the readers.

Thanks for your good comment. Our study teammates also wanted to have more extended time results. However, Taiwan government changed the vaccination policy and let our citizens choose any kind of vaccines and different vaccination strategies as they wanted at the end of 2021. Under the circumstances, long term follow-up and data collection became very difficult and inconsistent. So, we are very sorry we don’t have extended time results to share with our readers.

You can also see the attachment.

Round 2

Reviewer 1 Report

Authors satisfied all my concerns and suggestions, therefore I recommend the current version of manuscript for the publication.